# The Influence of Socio-Demographic Factors on the Forms of Leisure for the Students at the Faculty of Physical Education and Sports

**DOI:** 10.3390/ijerph182312577

**Published:** 2021-11-29

**Authors:** George Danut Mocanu, Gabriel Murariu, Dan Munteanu

**Affiliations:** 1Department of Individual Sports and Physical Therapy, Faculty of Physical Education and Sport, “Dunarea de Jos” University of Galati, 800008 Galati, Romania; george.mocanu@ugal.ro; 2Chemistry, Physics and Environment Department, Faculty of Sciences and Environment, “Dunarea de Jos” University of Galati, 800008 Galati, Romania; 3Department of Computers and Information Technology, Faculty of Automation, Computer Sciences, Electronics and Electrical Engineering, “Dunarea de Jos” University of Galati, 800008 Galati, Romania; dan.munteanu@ugal.ro

**Keywords:** opinions, questionnaire, students, active and passive leisure, favorite activities

## Abstract

The study investigates the influences of gender, area of origin and age stage variables and also of the interaction between them, on the free time behavior of the students at the Faculty of Physical Education and Sports from Galati. The questionnaire applied in the academic year 2019–2020 had 85 items and was structured on 4 factors: leisure budget, leisure limiting factors, preferred leisure activities, and leisure sports activities. The multivariate/MANOVA analysis showed statistically significant data for some of the analyzed items, with values of F associated with thresholds *p* < 0.05. The results support longer screen time for urban areas and for those <25 years and time limitation for the favorite activities of students >25 years, while reading had higher stress scores for men and students <25 years. Men tended to limit their free time working overtime and women limited their free time due to housework. Students from rural areas and men >25 years were more stressed by socializing on the internet and shopping. Financial limitations for preferred activities were higher for women and students <25 years—women read more and visited their friends more often while men had higher scores in relation to involvement in physical activities throughout the week, an aspect also reported for those <25 years. Students >25 years spent more time with their family, while those <25 years socialized more on the internet and had better scores when going out with friends. Those in urban areas did more jogging, men had better scores in relation to playing sports games, higher indicators for the satisfaction generated by sports activity, and women preferred jogging and cycling/rollerblading. Sports games and different types of fitness were the most common variants practiced at the level of the studied group. Conclusion: There was no dominant orientation of the investigated group towards forms of passive leisure and there were no cases of sedentariness, even if the use of technologies (video games, socializing on the Internet and TV) were forms of leisure often used by students.

## 1. Introduction

Leisure activities have a number of important benefits—they allow and develop autonomous behavior, create conditions for asserting one’s identity, for living and for solving new situations/contexts, with multiple positive effects in personal evolution [1]. Urbanization processes and social changes in recent decades have led to a decrease in the time spent on outdoor leisure activities for the younger generations, who are increasingly sedentary [2].

Working adults generally prefer passive leisure activities (such as watching TV), but this affects their performance and well-being. High-intensity leisure physical activities allow their detachment from work-related problems, call for major functions (physiological component) and generate increased subjective well-being and a high level of satisfaction (life satisfaction and subjective well-being) in the opinion of the authors of [3]. The sedentary behavior of young people (8–19 years) is negatively associated with academic performance for both sexes, and restrictions on sedentary activities (type of screen-based sedentary behavior) are beneficial for them [4]. The study by the authors of [5] indicated that sedentary adolescent boys are a high-risk group as they are most likely to be inactive for the rest of their lives. Passive leisure was at the top of the list for students at the University of Huelva/Spain with preferences for music, computers, going out with friends and TV. The least represented were trips, hikes, participation in conferences, sports activities and going to the disco. Men had better scores for leisure and physical activities and the use of video games, as well as a higher level of self-perceived health, according to the authors of [6].

Sedentary behavior and a high prevalence of physical inactivity for Southeast Asian countries was identified by the authors of [7,8]. Physical inactivity was associated with not walking or cycling to school, absence from physical education classes and poor support from friends and parents. Sedentary behavior was associated with older age, a high average income, a high BMI, loneliness, alcohol consumption and lack of parental supervision. In the same area, students with more than 3 h of sedentary leisure showed psychological suffering, those with more than 5 h consumed more tobacco, and those with more than one hour were prone to alcohol consumption.

The role of leisure and physical activities in alleviating stress states for students in Norway (15 years) was investigated by the authors of [9]. They found associations between low levels of physical activity and the manifestation of health problems, including high values of stress. The investigation carried out by the authors of [10] demonstrated that health sciences students had a better mental state than others/non-health science students. Physical activity, male gender and resilience are the predictors of positive mental health. Leisure time influenced the self-rated health of Taiwanese students. This was determined by good exercise habits, optimal BMI values and healthy eating habits, according to the authors of [11]. Holidays and tourism are a good way to reduce stress, increase perceived immune capacity and improve mood and mental health [12,13,14].

The importance of adolescence in the formation of stereotypes/lifelong habitual habits of physical activity was highlighted by the authors of [15]. The analysis of a group of adolescents from Porto (13–17 years old) showed that physical activity at the age of 13 was associated with a strong and stable involvement in physical activities at 17 years old. High PA levels were maintained more for boys (41.1%) than for girls (19.5%). Out of the sedentary 13-year-olds, 33% of girls and 32.3% of boys maintained this harmful lifestyle up to 17 years of age. The authors also pointed out that the urban environment in the vicinity of the area of residence did not affect the changes in PA in the free time of those investigated.

The teaching strategy based on the development of student autonomy (11–18 years, with a low educational level) in Germany and the generation of positive emotions during physical education lessons can favorably influence leisure physical activities throughout life, with beneficial effects on health [16]. Physical education teachers, along with the models offered by parents have a major and significant role in supporting leisure and relaxation activities for teenagers. However, the self-determining motivation of young people is also very important in shaping the favorable behavior aimed at engaging in leisure physical activities, according to the authors of [17]. The transfer of autonomous behavior and motivation for physical effort from physical education lessons to active leisure activities has been studied by the authors of [18], who insisted on the capitalization of the skills and competencies developed at school in daily life. The participation of young Norwegians in leisure activities could be promoted by the school, where gender differences and those related to socio-economic status can be blurred, thus generating a neutrality of leisure types, and girls and boys can become involved together in a diversified range of activities. It is very important to motivate teenagers to become involved in leisure activities where they can obtain favorable results, which would reduce or eliminate cases of early school leaving. In the case of adults, involvement in various activities is linked to prestige, higher qualifications and success, by making a selection of participants in relation to their competence according to the authors of [19,20]. The motivation and autonomy for leisure physical activities among Spanish adolescents was determined by the influence of parents, the physical education teacher and colleagues [21].

Organized physical activities for young people in Poland are important for ensuring high levels of vigorous PA (physical activity) while subjects who are not involved in organized physical activities had an increased risk of health problems. PA school programs must take into account the preferences, qualities, talent and individual weaknesses of students, according to the authors of [22]. The rational use of free time defines the development and well-being of adolescents in U.S. colleges, where sports, parties and various activities organized for young people are dominant, according to the authors of [23]. The authors highlighted the need for adults to structure and supervise the involvement of young people in leisure activities.

The relationship between sedentary or active lifestyle/walking within the residential area and the socio-economic status of the neighborhood for universities in Valencia/Spain was analyzed by the authors of [24]. Students living in more walkable areas had more commuting/walking activities to and from university to their home, while commuting was also present at a high level for LSS areas/areas defined by low socioeconomic status and the time spent on sedentary activities was longer for those in the SES areas.

The diversification of leisure sports physical activities for young people will generate high levels of physical activity in the next stages of adult life, a phenomenon found especially among women, according to the authors of [25]. Participating in various sports activities will generate pleasure and the development of motor skills, which will be used later for physical leisure options; sports clubs have the role of providing cheap and affordable options for teenagers to avoid their abandonment or reluctant attitudes to physical effort. The internationalization of higher education requires organizations and professionals in the field of leisure to take into account cultural contexts, in order to ensure a good involvement and higher commitment of foreign students, according to the authors of [26].

In Romania, the concerns of the young generation are still oriented towards physical leisure activities at the level of 11–12 years old (first grades in gymnasium), but with increasing age, the interests for leisure activities based on the use of technologies become dominant, contradicting the rules that define a harmonious, balanced and physically and mentally healthy lifestyle [27]. The opinion of high school students in Romania towards physical activities organized in school and leisure physical activities was investigated by the author of [28] and only 13% of the subjects investigated emphasized the need to exercise with their family in support of these activities.

The main factors that ensure a behavior based on physical activities in the situation in which the transition from adolescence to adulthood happens were identified by the authors of [29]. For women, the existence of sports skills perceived in childhood and adolescence was positively associated with the persistence of active behavior in adulthood, while smoking and the existence of younger siblings generated negative associations. For men, active behavior in adulthood was positively associated with the existence of physically active parents (as role models) and with the practice of sports activities outside the school curriculum.

In Norway, a study of 3251 young people (under the age of 19) made by the authors of [30] identified the predictors for a behavior based on leisure time physical activity (LTPA) or its limitations. High levels of LTPA in adults could be predicted by the membership/member of a sports club and the image of positive athletic self-concept in adolescence. In contrast, low levels of LTPA in adults were indicated by a high BMI/obesity, smoking and the onset of depressive symptoms in adolescence.

The study of leisure behavior for adolescents in the USA (Minnesota) identified a decrease in physical activity (PA) compared to the recommended values for this age stage. The factors that contributed to the involvement of young people in leisure physical activities were complex and partially overlap for both sexes. The personal context, the social context and the importance of PA in the school environment were relevant as well. Moreover, involvement in moderate and vigorous physical activity depended on the peer pressure/friends and family climate, so working with young people, friends and their families can promote PA, according to the authors of [31].

The analysis of the lifestyle of young people in urban and rural areas (189 cases) in Serbia (Leskovac) indicated that physical activity was present at a satisfactory level for children and their families [32]. Unstructured, passive and activities without a positive outcome (excessive media consumption) must be eliminated/neutralized by presenting the benefits of active leisure on health, because students are often willing to choose video games and computers at the expense of physical exercise, regardless of background. Favorable models offered by parents to children (physical activity) will generate the copying of these behaviors and a superior involvement of children in leisure sports physical activities. Students in urban areas are more often involved in structured activities outside of school, and those in rural areas are especially involved in structured activities organized by the school.

*The purpose of the study* was to measure the opinions of the students of the Faculty of Physical Education and Sports regarding the leisure activities, through a questionnaire containing 85 items and sub-items with closed answers, highlighting the common aspects and the differences signaled in other specialized studies.


*Working hypotheses:*


**Hypothesis** **1.** **(H1)**
*The questionnaire applied for the investigation of the target group accurately measures the features related to each analyzed factor, an aspect validated by the significant values of the internal consistency coefficient (Cronbach alpha).*


**Hypothesis** **2.** **(H2)**
*The average scores of the items of the questionnaire are not equal at the level of each factor of the questionnaire used.*


**Hypothesis** **3.** **(H3)**
*There are significant differences between the average values of the questionnaire items for the gender independent variable.*


**Hypothesis** **4.** **(H4)**
*There are significant differences between the average values of the questionnaire items for the independent variable area variables.*


**Hypothesis** **5.** **(H5)**
*There are significant differences between the average values of the questionnaire items for the independent variable age stages.*


## 2. Materials and Methods

### 2.1. Participants

The initial group included 224 students from the Faculty of Physical Education and Sports in Galati (undergraduate studies), made up on the basis of randomized selection, so that each of the 3 independent variables investigated (gender, area variables and age stages) would be represented in the final sample. Of these, 180 students remained in the study and provided feedback on the request to participate in this investigation (representing 40.17% of the total number of students enrolled in undergraduate studies at the time of the study, being a representative percentage for this specialization). A total of 44 students were eliminated from the data processing due to incomplete answers for some items, or a failure to honor the invitation to complete and return the questionnaire. The average age of the investigated group was 22.34 ± 5.79 years. A total of 3 independent variables were defined: gender (104 men and 76 women), area variables (120 urban and 60 rural) and age stages (150 < 25 years and 30 > 25 years). The subjects received and returned the questionnaires in electronic format by e-mail. The conducted research complied with the rules on voluntary participation and the confidentiality of the data collected and processed was ensured: the whole group were informed about the purpose of the research and treated in accordance with the Helsinki Declaration [33,34]. The study was conducted with the consent of the Ethics Commission of the higher education institution.

### 2.2. Procedure

The working methodology used was the usual one for attitudinal type investigations, given that the study was designed for cross-sectional research. The design and validation of the questionnaire was carried out within the Research Center for Human Performance within the Faculty of Physical Education and Sports, Galati. The questionnaire included 85 items and sub-items grouped on 4 distinctive factors: leisure budget (7 items), leisure limiting factors (28 items), preferred leisure activities (32 items) and leisure sports activities (18 items). The 5-step Likert scale (all answers are closed) was preferred to measure the intensity of the opinions expressed. Their quantification in scores, as well as the percentages obtained at the level of the whole batch for each item can be consulted in the appendix attached to this study. The large volume of data did not allow the presentation of scores and the intensity of opinions in the paper, but only a synthesis of the relevant percentages at the level of each factor. The variants for quantifying the intensity of opinions were selectively exemplified: 5 (Very much, All the time, Constant, Very strong influence), 4 (Much, Almost all the time, Often/frequent, Strong influence), 3 (Average, Half of the time, Occasionally, Medium influence), 2 (Little, Very little time, Rarely, Weak influence), 1 (Too little/Not at all, Scarcely, Never, Lack of influence). The questioning of students was performed in October–November of the first semester of the academic year 2019–2020 before the outbreak of the COVID-19 pandemic and the transfer to online teaching activities. The questionnaires are available online (Appendix A).

### 2.3. The Statistical Analysis of the Data

The statistical processing of the obtained results was performed using SPSS software version 24. The software calculated the percentage values of the scores expressed and related to each item for the whole batch, the values of Alpha Cronbach coefficients for determining the reliability value of the measurements of opinions expressed for each factor and the values of Hotelling’s T-Squared Test to verify the heterogeneity and variety of the expressed answers [35,36,37]. The study was based on determining the influence of the 3 independent variables (gender, area and age stages) and the interaction between them on the dependent variables represented by the answers to the questionnaire items. In this sense, the following were performed: multivariate/MANOVA analysis, identifying the effect/influence of each independent variable and their associations on each factor of the questionnaire, a univariate analysis to determine F and its significance thresholds at each level of the item separately, a size effect calculation materialized in partial eta squared values (η2p), a comparison of the averages of the scores expressed for each item at the level of each independent variable and the Bonferroni correction factor to determine the significance of the differences between the averages [38,39,40]. The confidence interval was set at 95% (*p* < 0.05).

## 3. Results

The large volume of resulting data makes it impossible to present them in full; for this reason the results were selected and distributed selectively in the univariate tables, but only the statistically significant ones. Table 1 summarizes the results of the internal consistency analysis, Tables 2, 4, 6, and 8 present the results of the multivariate tests analysis, and Tables 3, 5, 7, and 9 present those of the univariate tests analysis.

The values of the internal consistency coefficient (Cronbach’s Alpha) and the results of the equality test for the averages of the items of the 4 analyzed factors (Hotelling’s T-Squared Test) are presented in Table 1. For leisure budget and leisure sports activities factors, values of internal consistency >0.8 were obtained, and for leisure limiting factors and preferred leisure activities values >0.7, all were above the lower limit of 0.6, so the features measured by the questionnaire were measured faithfully, thus confirming the first hypothesis of the research/H1. Hotelling’s T-Squared Test indicated statistically significant results (*p* < 0.001), i.e., there was diversity in the selection/expression of item answers, the averages obtained were not equal and the aspect valid for all four factors was: F1 (6174) = 70.54; F2 (27,153) = 76.62; F3 (31,149) = 49.28; F4 (17,163) = 124.62, thus confirming the second working hypothesis/H2.

The influence of the three independent variables and the interaction between them on the answers to the items of factor 1/leisure budget is summarized in Table 2, with four reported situations of statistically significant results (*p* < 0.05).

For Area variables, an F = 2795 was obtained associated with a threshold *p* = 0.009, with 10.5% of the variance of the dependent variable attributed to the influence of the independent variable. The influence of Age stages generated an F = 5.375, associated with a threshold *p* = 0.000, and 18.5% of the variance of the items of this factor could be explained by the independent variable. The Gender* Age stages interaction obtained an F = 3.390, associated with a threshold *p* = 0.002, and 12.5% of the variance of factor 1 items was attributed to this interaction. The interaction Area variables * Age stages generated an F = 2.692, associated with a threshold *p* = 0.011, and 10.2% of the variance of the items could be explained by this interaction. For the independent variable Gender, the interaction Gender* Area variables and the interaction Gender* Area variables* Age stages were reported to have insignificant effects on the answers to the items of factor 1 (*p* > 0.05), as well as reduced values of size effect (η^2^_p_).

At the level of factor 1/leisure budget, the percentage values obtained by the whole lot are selectively presented for the items that compose it. Only 5.55% of students had over 5–6 h of free time daily, 27.22% had 4–5 h, 54.44% had 2–3 h, 9.44% had one hour and 3.33% had less than one hour. For favorite leisure activities, 62.22% allocated 2–3 h daily, 18.33% allocated 1–2 h, 8.33% allocated less than one hour, and 2.77% allocated over 5–6 h. For 54.44% of the students, free time was important, for 26.66% it was very important, and for 1.66% it was not very important. The use of screens (TV, computer games, internet) was allocated half the time by 53.33% of respondents, 28.88% only for basic information, 15% allocated almost all free time and only 2.77% were not interested in this option. Only 4.44% always left their place of residence on the weekends, 47.22% occasionally, 31.66% rarely and 3.88% never. On the other hand, 46.11% were medium satisfied with the way they managed their leisure time, 6.11% were very satisfied, 10% were not very satisfied and 1.66% were totally dissatisfied.

For the items of the leisure budget factor presented in Table 3, men obtained higher scores for almost all questions, except for the importance of leisure time, where women had a higher average value, but all pairs of the independent gender variable showed statistically insignificant differences (Values of F corresponded to thresholds *p* > 0.05). At the level of the variable area of origin, those in urban areas had higher average scores for Free time on working days, Free time for TV and internet (where it was the only significant difference/*p* = 0.031) and Satisfaction in organizing free time. Those in rural areas had higher average scores.

The importance of free time, Hours allocated to favorite activities and Weekends spent out of town, but without significant differences reported. At the level of the age stage variable, the most values of F associated with significant thresholds were obtained (*p* < 0.05). The group of students <25 years had higher values than the group >25 years for most items, except The importance of free time, where students >25 years had higher values, but the difference was statistically insignificant (*p* = 0.434), showing that time had a higher value with age. Significant thresholds were recorded for Leisure budget size and Free time on working days (the greater responsibilities of those over 25 limit them in these respects), and the younger ones had statistically higher and statistically significant scores for Hours allocated to favorite activities, but were concerned by the fact that they spent significantly more time in front of screens (TV, games, internet, etc.)/the Free time factor for TV and internet.

Table 4 indicates the results of the multivariate analysis at the level of the items of factor 2 leisure limiting factors. There were four cases of statistically significant influence (*p* < 0.05) of the independent variables and the combinations between them on the responses to the items of this factor. The variable Gender generated an F = 1.652, associated with a threshold *p* = 0.030, and 24.2% of the variance of the dependent variables (items of this factor) could be explained by the independent variable.

For Area variables, we obtained an F = 1.942, associated with a threshold *p* = 0.006, and 27.3% of the variance of the answers to the items was attributed to this independent variable. At the level of the Age stages variable, an F = 2.715 was registered, which corresponded to a threshold *p* = 0.000, with 34.4% of the variance of the obtained answers attributed to this independent variable. The interaction of Area variables* Age stages generated an F = 1.825, with a threshold *p* = 0.012, and 26.1% of the variance of the answers was determined by this combination of independent variables. For the rest of the interactions between variables (Gender* Area variables, Gender* Age stages and Gender* Area variables* Age stages) the values of F were associated with insignificant thresholds (*p* > 0.05), with low values of size effect/η^2^_p_ also found.

At the level of factor 2/leisure limiting factors, the percentage values for the items that define it were synthesized. The main reason that generated the loss of free time was household activities, with 7.22% of respondents allocating the code very much and 10% much. Working overtime indicated a very high score for 3.33% and a high score for 10.55%. Difficult homework also generated a loss of free time: 1.66% very much, 9.44% much and 55.55% an average value. The factor that least influenced the loss of free time was the commute, with 46.66% assigning the grade not at all, and 25% the grade little.

Among the leisure activities, high values of stress were recorded by watching TV and playing computer games (3.88% extreme stress, 5% strong stress, 16.66% medium stress, although 50% of students indicated the absence of stress). An unpleasant surprise was the stress generated by reading (2.77% extreme stress, 6.66% strong stress, 25% medium stress). Shopping obtained 1.66% for extreme stress, 6.66% for high stress and 28.33% for medium stress, and visits to relatives and friends generated 2.77% extreme stress, 6.11% strong stress and 16.66 medium stress. Stress-free activities were as follows: walks in the park (82.77%), going out with friends (77.77%), trips and hikes (76.11%), cinema, theater (75.55%) and listening to music (73.88%).

For preferred leisure activities, financial resources were decisive for 3.33% of students, were important for 19.44%, had a moderate importance for 54.44% and were irrelevant for 3.33% of them. The most accessible leisure activities from the financial point of view were walks in the park (84.44% very easy and 15% easily accessible), watching TV and computer games (78.33% very easy and 18.88% easily accessible), socializing on the internet (65% very easy and 31.66% easily accessible), listening to music (58.33% very easy and 32.77% easily accessible), visits to relatives and friends (58.88% very easy and 30% easily accessible). The most financially inaccessible were trips and hiking (3.88% inaccessible and 30.55% hard accessibility), shopping (17.22% hard accessibility and 57.66% medium accessible), going out with friends in the city (5.55% hard accessibility and 54.44% medium accessible), cinema and theater (2.11% inaccessible, 6.11% hard accessibility and 47.77% medium accessible). Sports activities obtained scores of 6.66% inaccessible, 20% medium accessible, 55% easily accessible and 18.33% very easy. The explanation for the fact that they were not on top was the fact that certain tests and subjects practiced involve considerable financial investments for food, equipment, rent, etc.

Table 5 selectively presents the most relevant results from the analysis of the univariate tests for factor 2/(leisure limiting factors), the average values for the established pairs of independent variables and the statistical significance of the difference between them. Men obtained higher scores associated with sacrificing time due to working overtime and commuting, and women had higher scores for difficult assignments, helping others and household activities. For the latter factor a significant difference between the sexes was recorded (*p* = 0.003).

Men obtained higher scores for the stress induced by leisure activities than women for almost all the variants expressed, except for socializing on the internet, where women had a slightly higher value, but the difference between the sexes was insignificant in this case. Even if the average values were between 1 (no stress) and 2 (low stress) a number of significant differences were reported for the following dependent variables: going out with friends (*p* = 0.035), reading (*p* = 0.029), listening to music (*p* = 0.001) and walks in the park (*p* = 0.027), so men were more stressed primarily by the passive leisure options, but also by the forms of the active ones. In general, women felt more financially constrained to their favorite activities, but the difference compared to men was statistically insignificant in all cases. Men had higher scores for financial inaccessibility for going out with friends in the city and visiting relatives/friends, and women obtained higher scores for all other activities.

Difficult homework and helping others were the factors where urban students had higher average scores, and overtime work, housework and commuting generated higher average scores for those in rural areas. In the latter case, the difference was significant (*p* = 0.000). Rural students had higher perceived stress scores for all leisure activities, with significant differences for various sports activities (*p* = 0.007), walks in the park (*p* = 0.005) and socializing on the internet (*p* = 0.002). The financial limitations for leisure activities were more pronounced for those in rural areas but the differences compared to urban were all insignificant (*p* > 0.05).

Limiting time by Working overtime (*p* = 0.022), household activities (*p* = 0.004) and commuting (*p* = 0.008) generated higher scores for students >25 years and implicitly significant differences from the group of <25 years, which in turn had higher values for the time lost with difficult homework and help given to others, but the values of F were associated with thresholds of insignificant differences in these cases. Reading was the only activity that was significantly more stressful for those <25 years, compared to those >25 years (*p* = 0.036), demonstrating the increasingly weak reading concerns of young people. All other activities induced higher average stress values for subjects >25 years, with significant differences for walks in the park (*p* = 0.004), socializing on the internet (*p* = 0.000) and shopping (*p* = 0.043). Financial limitation of preferred activities obtained a higher average score for those <25 years (whose incomes are certainly lower), but the difference between groups was statistically insignificant, as were the differences between the average scores for the financial accessibility of all forms of leisure.

The influence of the independent variables and the interaction between them on the responses to the items of factor 3/preferred leisure activities is summarized in Table 6, but there is only one situation in which there were statistically significant results (*p* < 0.05). For Age stages, an F = 1.901 was obtained, associated with a threshold *p* = 0.006, with 30.1% of the variance of the answers to the items of the questionnaire attributed to the influence of this independent variable. For the other independent variables (Gender and Area variables), as in the case of all interactions between variables, statistically insignificant values of F (*p* > 0.05) were obtained, complemented by poor size effect results (η^2^_p_).

At the level of the factor 3/preferred leisure activities, the relevant percentage values for the items that compose it are summarized. Leisure indicated high scores for family members (18.33% constant, 30% frequent and 35.55% occasionally), schoolmates/peers (6.11% constant, 38.33% frequent and 37.22% occasionally) and life partner (27.22% constant, 30% frequent and 13.33% occasional). Low scores were obtained for the variants: pet (6.66% constant, 17.22% occasionally and 52.22% never) and for the option alone (4.44% constant, 20% occasionally and 23.88% never).

The forms of leisure in which students engaged during the week focused primarily on sports activities (17.77% constant, 40% frequent, 27.22% occasionally) and socializing on the Internet (17.22% constant, 41.11% frequently, 26.11% occasionally), so those two forms of active and passive leisure were not mutually exclusive due to the specialization of the students in the group. Other high scores were obtained for listening to music (12.77% constant, 16.11% frequent, 31.66% occasional), TV and computer games (5.55% constant, 15.55% frequent, 32.78% occasional), going out with friends in the city (3.88% constant, 21.66% frequent, 48.88% occasional), shopping (1.66% constant, 16.11% frequent, 43.33% occasional) and walks in the park (1.66% constant, 16.66% frequent, 40% occasional). The weakest involvement was for trips and hikes (47.78% never, 46.66% rarely, 5% occasionally), cinema/theater (26.11% never, 53.33% rarely, 18.33% occasionally), visits to relatives, friends (15.55% never, 51.66% rarely, 25% occasionally) and reading (18.33% never, 47.22% rarely, 25.55% occasionally and only 2.77% constantly), the last aspect demonstrating the increasingly weak concern of young people for books/fiction. For the weekend variants, increases were observed for most activities but unfortunately the only decrease was for sports activities, which still remained on top (11.67% constant, 33.33% frequently, 36.37% occasionally) but was exceeded as a basic concern by socialization on the Internet (20% constant, 38.89% frequently, 25% occasionally).

For the holidays, there was a balance related to the internal variants at sea (16.67% always, 25.55% often, 30% occasionally) and in the mountains 14.44% always, 22.22% often, 36.11% occasionally). Holidays in the countryside were a well-represented option (11.11% always, 21.67% often, 22.22% occasionally) and spending holidays at home could be associated with the lower financial strength of the group studied (7.22% always, 37.22% often, 27.22% occasionally), strengthened by the weak percentages allocated to holidays abroad (6.11% always, 5% often, 18.33% occasionally, 33.89% rarely, 36.67% never).

Table 7 summarizes the results of the analysis of variance and the differences between the average values of the scores of the items of factor 3/preferred leisure activities, at the level of the pairs of the three independent variables. Women had higher average scores for spending time with their family and life partner, and men tended to spend more time with peers, pets or alone, but all gender differences were statistically insignificant (*p* > 0.05). Men obtained higher scores of leisure time during the week for going out with friends in the city, socializing on the internet and various sports activities. The latter had a significant difference (*p* = 0.007). Women had higher scores on all other activities, with significant differences in reading (*p* = 0.029) and visits to relatives/friends (*p* = 0.033). At the weekend the situation was similar, with significant differences in physical activities for men (*p* = 0.007) and differences in favor of women for reading (*p* = 0.048). Men had a higher score for holidays spent at home, and women for all other options, but all differences were statistically insignificant (*p* > 0.05).

Subjects from rural areas had higher scores in relation spending time with family and friends/peers, while urban subjects had higher scores in spending time with their life partner, pet, and alone, with no statistically significant differences reported. Going out with friends, reading, visiting relatives/friends and excursions/hiking were the items where those in the countryside had superior results in relation to spending their free time during the week, and at weekends they had higher scores only for visits of relatives/friends and trips/hiking, the rest of the variants having higher average values for urban students, but all the differences between the pairs were insignificant. For holidays, students in urban areas had better scores for holidays abroad, holidays in the mountains and at the seaside, where the only significant difference was reported (*p* = 0.048). Those in rural areas had higher scores for spending holidays in the countryside/relatives/grandparents and for those spent at home.

Students aged >25 years had higher average scores for spending free time with their life partner and their family, respectively, where a significant difference was also reported (*p* = 0.000). Those in the group <25 years had higher average scores for the time spent alone, with their pet and or peers and in the latter case the difference was significant (*p* = 0.000). Those aged >25 had higher average values for reading, visits to relatives, trips and hiking, cinema/theater for the activities during the week, for the rest being reported higher values of those aged <25 years, with the existence of three significant differences: going out with friends in the city (*p* = 0.030), various sports activities (*p* = 0.008) and socializing on the internet (*p* = 0.001), so we can say that younger students have a more dynamic lifestyle and socialize more easily, but they are even more dependent on technology. For weekend activities, those >25 years old maintained their superiority only for reading and excursions/hiking, and those <25 years old obtained significant differences for going out with friends in the city (*p* = 0.001) and socializing on the internet (*p* = 0.004). Students aged >25 obtained higher average scores for vacations in the mountains and abroad (one reason would be better financial status, the fact that many have jobs) and those in the group <25 preferred more holidays at sea and the other variants, without any significant differences between age groups.

Table 8 summarizes the results of the multivariate analysis at the level of factor 4 of the questionnaire/leisure sports activities. Significant values of the influence on the answers in the questionnaire (*p* < 0.05) were obtained only for the Gender variable, with F = 2.288, a result associated with a *p* = 0.003, and 21% of the variance of the answers (dependent variables) was determined by this independent variable. The other two independent variables (Area variables and Age stages) as well as the four variants of interactions between the variables generated insignificant results of F (*p* > 0.05), as well as weak values of η^2^_p_.

As expected, for factor 4/sports leisure, percentages were obtained that demonstrate that most of the investigated students had a dynamic/active lifestyle. Not even one subject was sedentary, and 15% thought they had a dynamic lifestyle, 42.22% frequently exercised, 42.22% had moderate involvement in physical activities. All respondents were involved in indoor and/or outdoor physical activities: 18.33% constantly, 38.89% often/4–5 days a week, 27.22% moderately/2–3 days a week, 15.55% rarely. Physical effort generated 20% of the cases of strong satisfaction and fulfillment, for 57.78% well-being and high tonus, for 17.22% moderate comfort and only for 5% a reduced state of comfort.

The top physical activities of interest included sports games (18.33% constant practice, 18.33% frequent practice, 34.44% occasional practice), fitness/bodybuilding activities (9.44% constant practice, 21.11% frequent practice, 25% occasional practice), jogging (5.55% constant practice, 13.33% frequent practice, 46.67% occasional practice) and not at all surprising—due to the rise in recent years—cycling/rollerblading (2.22% constant practice, 9.44% frequent practice, 45.56% occasional practice). The lowest scores were obtained for contact sports/boxing, karate, wrestling (74.44% of students did not practice them) and unfortunately swimming (44.44% of students never practiced it), but also tennis/table tennis (36.11 % of students did not practice them).

Swimming practice had the strongest effects on body harmony and health (31.67% very strong influence, 52.22% strong influence, 12.22% average influence), even if it was not the most practiced/deficient logistics would be the explanation. This was followed by activities in fitness/bodybuilding (25% very strong influence, 39.44% strong influence, 28.89% average influence), jogging (15.56% very strong influence, 34.44% strong influence, 43.33% average influence) and sports games (15% very strong influence, 46.67% strong influence, 32.22% medium influence). The weakest scores were attributed to tennis (40.55% weak influence, 5% lack of influence), contact sports (22.22% weak influence, 8.89% lack of influence—perhaps through injury risks) and cycling and rollerblading (20% weak influence, 7.72% lack of influence).

Table 9 selectively indicates the results of the univariate analysis for the items of factor 4/sports leisure and the main average scores associated with each item. Men were more involved in physical activities, they had a sportier lifestyle, physical activities were more important to them. Significant differences were reported between the satisfaction generated by the physical effort—higher also in men, with a value of F associated with a threshold (*p* = 0.038). Regarding the sports/tests, women had a higher score only for jogging and cycling/rollerblading and in the second case a significant difference was reported (*p* = 0.047). For the other variants, the superiority of the average scores of men is noticeable, obtaining significant differences in the case of sports games (*p* = 0.002) and tennis/table tennis (*p* = 0.002). Regarding the favorable effects on health and body harmony, no significant differences between the sexes were reported; women had higher average scores for jogging, tennis and cycling/rollerblading, and men for the rest of the variants.

Students in urban areas had higher scores related to active lifestyle, involvement in physical activities and satisfaction produced by physical effort, without reporting significant differences from those in rural areas. Those in urban areas practiced more jogging, with a significant difference (*p* = 0.034), fitness/bodybuilding, swimming and cycling/rollerblading those in rural areas the rest of the activities, but without statistically significant differences. With the exception of tennis, urban students had higher average scores for the rest of the sports than those in rural areas, but no significant differences were reported.

The group of those aged <25 years had a more active lifestyle and the difference in this case was significant (*p* = 0.023). They had higher scores and involvement in sports activities and the level of satisfaction generated by effort, so advancing in age reduces participation in physical activities. Subjects aged >25 years practiced swimming more, for the other varieties of sports activities with lower scores compared to younger students, but without registering statistically significant differences. Sports games, fitness/bodybuilding and contact/boxing sports, karate, wrestling contribute more to body harmony and health optimization in the vision of the students <25 years, but even here no significant differences were reported. Table 10 summarizes the essential results by factors for the undertaken study.

## 4. Discussion

The studies that approach the topic of leisure activities are diverse. Their results are largely in agreement with our research, but there were also some aspects that were not confirmed. A synthesis of the selected specialized sources from 47 countries on the main continents, in connection with the forms of leisure practiced has been made by the authors [41]. Walking, running, and playing football/soccer were the main leisure activities, and along with swimming were also the most financially accessible sports leisure activities compared to other team sports (aspects similar to our study, except swimming, which was less practiced, especially by girls). Walking dominates the Western Pacific (41.8%), Southeast Asia (32.3%), America (18.9%) and the Eastern Mediterranean (15%). In Europe and Africa, football was the top activity (10%), followed by running (9.3%)—an aspect confirmed by the orientation of the young students in our study, but the options of teenagers still differ depending on the specifics of the analyzed region.

The analysis of leisure activities for young people in the Czech Republic (13–15 years old) identified how their participation in OLTA (organized leisure-time activities) contributes to reducing health risk behaviors (smoking, alcohol consumption, early sexual intercourse, etc.) and influences school/academic performance [42,43]. Those who did not participate in OLTA were more prone to the manifestation of these risky behaviors, to their involvement in aggressive/combat activities and implicitly to hurt themselves. Among girls, there were lower values for the manifestation of unwanted behaviors and school dropouts. Young Italians who participate in organized activities/out of school leisure time activity have lower rates of smoking, alcohol consumption and favorable impact on health behaviors, high life satisfaction, are more involved in physical activities [44].

Ways of spending free time for high school students in Hungary were identified by the author of [45]. They had average values during the week of 3.6 h/day and 6.6 h/day on weekends, and 87% did regular physical activity, without significant differences between the sexes. (For the group in our study, 62% of respondents allocated between 2 and 3 h of daily leisure activities, and only 2.77% had more than 5–6 h of free time per day). However, boys in particular spent more free time on TV or computers, being involved in video games (fighting games) and sports (e-sports), which reduced the free time allocated to physical activities, an aspect that was not confirmed by our investigation. These aspects, combined with the consumption of carbonated drinks and fast food were factors that predispose to the occurrence of chronic diseases.

Excessive use of the Internet at work and as a form of leisure can be a risk factor for mental health. Those who use the internet moderately/without excesses have a higher perceived quality of life, according to the authors of [46]. The association between sedentary behavior, the time spent in front of the screen/screen time and the duration of short/long sleep, with implications on the manifestation of obesity in students in Zagreb/Croatia is highlighted by the authors of [47].

The main active and sedentary behaviors for young Scots were studied by the authors of [48,49]. Watching TV programs was the biggest consumer of free time, with about 33% of the total amount of free time, a lower percentage of teenagers allocating TV over 4 h a day, especially on weekends. Other identified sedentary activities refer to solving homework, involvement in video games or computer games (aspects similar to our study) and motorized transport. These activities had average values of 244 min. for girls and 228 min. for boys during the week, respectively 400 min. for girls and 326 min. for boys on weekends. In our study, 53.3% of students spent half of their free time on TV, computer games and socializing on the Internet, but there were no cases of full-time allocation for these variants of passive leisure.

Women had a lower level of physical activity, (similar to our research), and students in the Visegrad countries indicated higher involvement in physical effort than those in Ukraine, who, however, have better BMI levels, according to the authors of [50].

A study analyzing the values of free time and the efficiency of its use for young people in recent decades was conducted by the author of [51]. He notes that they have more and more free time available and allocate less and less time to paid activities. The 2003–2005 interval indicated an average daily value of 6 free hours, but 2/3 of this time was intended for passive leisure activities and only 1/3 was oriented towards activities that facilitate the formation of skills and can contribute to personal growth.

The analysis of the dominant leisure options for adolescents in Portugal (active and sedentary) indicated a greater inclination of boys for sports activities, but also a greater involvement in the use of computers and TV (similar to our study), and girls obtained high scores for individual activities, artistic and social leisure. Both sexes showed significant associations between the level of PA and social leisure, according to the authors of [52]. A study to determine the share of free time allocated to watching TV programs by young Brazilians (13–18 years) in the city of Aracaju showed that girls (70.9%) and boys (66.2%) spent more than 2 h a day for TV, and among them are predominantly girls with low socio-economic status, those with inadequate fruit consumption and boys over 16 years of age. It was recommended to replace this form of passive leisure with useful and productive activities, according to the authors of [53]. Over 23% of high school students in Lima/Peru watched TV for more than 2 h a day, and their favorite activities were TV, video games, and the Internet, which, when overused, were associated with poor school results [54].

The research conducted by the authors of [55] a decade ago on adolescents in three regions of Spain did not signal significant differences between active and sedentary subjects, in terms of the amount of time spent watching TV, computer games and solving homework. Those who were active only devoted more time to leisure physical activities, and their age and place of residence were decisive for the amount of physical exercise and the way they spend their free time. The idea was reinforced in his study in [56], which did not find significant longitudinal associations between watching TV programs and changes in leisure physical activity, with moderate and vigorous intensity, for young people (10–15 years). The two activities (active leisure and watching TV) are not functional opposites, the simple restriction of watching TV will not necessarily increase the level of involvement in physical activity (PA). Our research confirms these statements, at the level of the studied group being present both physical activities and variants focused on TV, internet, computer games, which are not mutually exclusive.

A common variant of leisure found by the authors of [57] was the use of video games (for cases aged 11–15), with much higher values for boys than girls. A lack of mediation/poor parental involvement was found in cases that exceeded 2 h/day, and cases of concern exceeded 4 h daily. The use of video games on the Internet affects the quality of life. A study conducted by the authors of [58] on vocational training students—who had used these games for a year—indicated low values of quality of life/poorer health related quality of life. Instead, the study by the authors of [59] indicated that video games and active motor games were effective in the field of physical education, motivating university students to have an active lifestyle.

The differences in leisure time for a group of Latin American and African American adolescents in urban areas (average age of 16.6 years), all with low-income families were highlighted by the authors of [60]. Their division and study on three different interest groups showed that subjects in the group with academic concerns had the highest levels of adjustment in all areas, those in the social cluster group had the poorest school results and high level of problematic behaviors, and those in the group of computer/TV users had the weakest values of intrinsic motivation.

Research on the perceptions of Portuguese adolescents (differences of opinion between the sexes) and how natural and social environmental factors (quality of walking and cycling routes, distance from locations for sports activities, aesthetics and pleasure generated by the study environment and family, connectivity and network quality) influence involvement in leisure activities was carried out by the authors of [61]. There was a greater involvement of boys and higher intensities of effort in physical activities compared to girls (similar to our study), and the increased distance from the locations of leisure sports generated negative associations with involvement in physical effort for both sexes.

The importance of the quality of the environment in school neighborhoods for Canadian adolescents, in terms of their involvement in forms of physical/active leisure was studied by the authors of [62]. The authors found that the number of green spaces and parks placed at a distance of 750 m from schools were positively associated with the involvement of students in sports leisure activities, for both sexes/genders. The safety of the neighborhood environment (Czech Republic and Poland) is a factor that increases the chances of walking/walking for teenage girls, the safe environment generates values of 11,024 steps/day, the least safe only 9686 steps/day, according to the authors of [63].

The motivation of young women in Canada (14.3 years as an average age) for physical activity-related behavior is studied by the authors of [64]. The motivation for physical activities, initially developed in an organized manner (fitness club) can generate a good involvement in the PA of personal free time and in the following stages (over an interval of 3 weeks). The main reason why potential customers sign up and continue physical activities in sports centers is that oriented towards obtaining and maintaining a good physical shape (staying fit), in the vision of the authors of [65]. In our research, those involved in leisure physical activities were motivated by the beneficial effects of various sports activities in terms of health and body harmony, physical activity being a lifestyle for them.

The favorite leisure activities of medical students in Saudi Arabia are surfing social networks, movies, spending time with friends and family, using the internet (variants identified by us). Only 4.3% of the respondents have more than 30 min of physical activity daily, and women prefer social networks and movies more than men, according to [66].

The existence of associations between leisure physical activities with mental well-being and subjective health for Finnish adults (42–50 years old) was verified by the authors of [67]. They determined that walking was positively associated with psychological and social well-being, activity in nature (rambling in nature) had positive associations with emotional and social well-being, and endurance training with subjective health, so the variants of free time physical activities are related to mental well-being and subjective health in middle age. Our study also signaled mental satisfaction and inner comfort produced by physical activities.

Problems in meeting the PA standards recommended by the WHO (World Health Organization) for undergraduate students in Madrid were reported by the authors of [68], 55.6% of these did not meet LTPA (leisure time physical activity) standards. Leisure time sporting activity for young people in Germany (1008 cases with an average age of over 14 years) gave practitioners high levels of general life satisfaction and specific activity (work, relationships, health, free time), according to the authors of [69]. The authors found that outdoor sports activities and those organized in clubs generated additional benefits, and the variety of sports leisure activities offered high degrees of satisfaction and contentment.

The spiritual dimension and religious life had greater associations with the involvement in physical activities and the pursuit of a balanced lifestyle for adolescents in the Czech Republic, according to the authors of [70]. Those involved in religious activities, who have spiritual concerns are less tempted to use the Internet excessively, being more often involved in organized activities and various forms of leisure, they read more and regularly, are likely to play a musical instrument, watch less TV and spend less time playing computer games, so the spiritual dimension has greater associations with involvement in physical activities. The study by the authors of [71] revealed that adolescents (15–18 years) with a higher level of involvement in various social activities have fewer mental problems, obtaining low scores associated with suicidal ideas. In the case of our group, older age was associated with a lower involvement in screening options (TV, games, socializing on the Internet).

Predictors for involvement in moderate and vigorous leisure physical activities (at the level of young people aged 14–18/high school) are the male gender, the index of relative autonomy and higher social participation. Participation in physical activities and an active lifestyle were associated with environmental factors, higher neighborhood safety, large family social capital, social participation and motivation for physical activities [72]. For our group, some of these aspects were also confirmed, especially for the participation of men in physical activities and better socialization.

The investigation made by the authors of [73] at the level of young people in Australia (10–16 years old) demonstrated that school physical activities and leisure activities, participation in representative teams of educational institutions were associated with reduced levels of stress and depression, combined with reduced time spent in front of screens (leisure time screen). The importance of physical activity in reducing depression for students in China in the first year of study is mentioned by the authors of [74], the physically active have high motivation to participate in leisure physical activities and reduced depressive symptoms compared to sedentary people.

## 5. Conclusions

The study facilitates a deeper understanding of the personality, needs and problems of the investigated students, identifying important features of lifestyle, expectations and areas of interest for them and their motivation for different leisure options. At the level of the multivariate analysis, the age stages variable was the one that generated significant influences (F values correspond to thresholds *p* < 0.05) for three factors (leisure budget, leisure limiting factors, preferred leisure activities), followed by the gender variable, for two factors (leisure limiting factors and leisure sports activities) and the area variable respectively—still for two factors (leisure budget and leisure limiting factors).

Working hypotheses 3–5 were only partially confirmed. There were numerous items where the differences in the pairs of independent variables or the associations between them were not statistically significant (The values of F in the case of univariate tests correspond to thresholds >0.05). We can say that there was no dominant orientation of the studied group towards forms of passive leisure, mentioned in some of the analyzed sources. A positive aspect was the lack of sedentary students, a phenomenon that can be explained by the nature of the university specialization of the investigated group.

At the level of factor 1 (leisure budget), there was a longer screen time, allocated by those in urban areas and those <25 years, less free time allocated to favorite activities for those >25 years, as a result of changes associated with a different lifestyle and changing priorities. The fact that screen time (TV, computer games, internet) was allocated over 50% of the daily free time budget demonstrates the dependence of the young generation on technology and it is a defining feature of young people’s behavior, according to the analyzed research.

At the level of factor 2 (*leisure limiting factors*) it is noted that men wasted more time working overtime, women and those >25 years with household activities, and the level of stress generated by going out in the city was higher for men. The financial factor is a limitation of involvement in favorite activities more for women and younger students (<25 years). Reading was perceived as more stressful for male students and younger ones, due to the orientation towards other concerns facilitated by technological progress (computer games), an aspect that contrasts strongly with the lifestyle of previous generations.

At the level of factor 3 (*preferred leisure activities*) it was observed that younger students (<25 years) tended to socialize more easily on the internet, go out more with friends, but also do more sports, thus signaling a balanced combination between forms of active leisure and variants of the passive leisure. There was a tendency of students >25 years to spend more time with family, but also the fact that men have higher scores of involvement in the forms of active leisure all week, thus having an inclination towards a more dynamic life.

For factor 4 (*leisure sports activities*) it was noted that men had a higher level of satisfaction generated by physical effort. They had a dominant focus on sports games, and women obtained higher scores for jogging and cycling/rollerblading (which does not involve high-level muscular and functional demands). It should be stressed that sports games and forms of fitness had the highest practice scores at the level of the studied group, and combat sports, tennis and swimming had the lowest participation rates, which draws attention to the problems of promotion and poor infrastructure at a local level for these forms of sports leisure.

### Limits of the Study and Future Investigation Directions

There were also unsubstantiated aspects in this study, on which other research has focused, such as highlighting unhealthy behaviors (smoking, banned substances, alcoholism). The relatively small number of students and the fact that the investigated students belonged to a special/separate field in higher education did not allow a generalization of the results for the entire university population, while the other undergraduate fields should be investigated separately and then a comparison between these specializations should be made. A repetition of the study at the faculty level would be necessary, the opinions expressed being valid for the period of normality before the pandemic, the changes in the socio-economic context and the restrictions imposed during the pandemic certainly affecting the leisure options of the students. Another direction of research would be to study the leisure-related views of graduate students (masters and doctorates), in order to identify the changes induced by advancing in age and changing priorities in life. The comparison of leisure forms by age categories (young/adults/elderly) would identify the major changes that occur in ontogenesis in this regard. Identifying the associations between the time allocated to physical activities or passive leisure and the value of academic performance would be an interesting topic to analyze. Last but not least, the investigation of students’ opinion on the diversification of forms of leisure in fitness centers (Bag Boxing, Tae-Bo, Kangoo-Jumps, Hatha Yoga, Pilates, Zumba, Body Pump, Spinning, Step Aerobics, etc.) and determining the advantages and reasons for involvement in these forms of sports leisure could be another research direction.

## Figures and Tables

**Table 1 ijerph-18-12577-t001:** The values of the reliability value coefficient (Cronbach’s Alpha) and the test of equality of the averages for the answers to the items of the 4 factors of the questionnaire/N = 180.

Factors	Reliability Statistics			Hotelling’s T-Squared Test				
	Cronbach’s Alpha	Cronbach’s Alpha Based on Standardized Items	N of Items	Hotelling’s T-Squared	F	df1	df2	Sig.
**F1**/leisure budget	0.811	0.803	7	435.408	70.54	6	174	0.000
**F2**/leisure limiting factors	0.711	0.748	28	2420.580	76.62	27	153	0.000
**F3**/preferred leisure activities	0.728	0.750	32	1835.334	49.28	31	149	0.000
**F4**/leisure sports activities	0.815	0.812	18	2326.630	124.62	17	163	0.000

**Table 2 ijerph-18-12577-t002:** The results of the Multivariate Testsa (MANOVA)/F1(leisure budget).

Effect	λ	F	Hypothesis df	Error df	Sig.	η^2^_p_	Observed Power
Gender	0.936	1.616 ^b^	7.000	166.000	0.134	0.064	0.657
Area variables	0.895	2.795 ^b^	7.000	166.000	0.009	0.105	0.909
Age stages	0.815	5.375 ^b^	7.000	166.000	0.000	0.185	0.998
Gender* Area variables	0.948	1.310 ^b^	7.000	166.000	0.248	0.052	0.548
Gender* Age stages	0.875	3.390 ^b^	7.000	166.000	0.002	0.125	0.959
Area variables* Age stages	0.898	2.692 ^b^	7.000	166.000	0.011	0.102	0.896
Gender* Area variables* Age stages	0.935	1.644 ^b^	7.000	166.000	0.126	0.065	0.666

a. Design: Gender + Area variables + Age stages + Gender* Area variables + Gender* Age stages + Area variables* Age stages + Gender* Area variables* Age stages; b. Exact statistic; λ—Wilk’s lambda; F—Fisher test; df—degrees of freedom; Sig.—level of probability; η^2^_p_—partial eta squared.

**Table 3 ijerph-18-12577-t003:** The results of univariate tests/ANOVA and the comparison in pairs of the average values at the level of the items for factor 1 (leisure budget).

Dependent Variable	Group	Mean ± Std. Error	a–b	F (1172)	Sig. ^b^	η^2^_p_	Observed Power
F1.5 Free time for TV and internet	a. urban	2.701 ± 0.078	0.391 *	4.744	0.031	0.027	0.582
	b. rural	2.310 ± 0.162					
F1.1 Leisure budget size	a. <25 years	3.349 ± 0.069	0.718 *	11.810	0.001	0.064	0.927
	b. >25 years	2.631 ± 0.197					
F1.2 Free time on working days	a. <25 years	3.351 ± 0.068	0.824 *	15.856	0.000	0.084	0.977
	b. >25 years	2.527 ± 0.195					
F1.4 Hours allocated to favorite activities	a. <25 years	2.918 ± 0.067	0.777 *	14.835	0.000	0.079	0.969
	b. >25 years	2.141 ± 0.190					
F1.5 Free time for TV and internet	a. <25 years	2.918 ± 0.059	0.825 *	21.141	0.000	0.109	0.996
	b. >25 years	2.093 ± 0.169					

*. The mean difference is significant at the 0.05 level; ^b^. Adjustment for multiple comparisons: Bonferroni.

**Table 4 ijerph-18-12577-t004:** The results of the Multivariate Tests (MANOVA)/F2(leisure limiting factors).

Effect	λ	F	Hypothesis df	Error df	Sig.	η^2^_p_	Observed Power
Gender	0.758	1.652 ^b^	28.000	145.000	0.030	0.242	0.979
Area variables	0.727	1.942 ^b^	28.000	145.000	0.006	0.273	0.994
Age stages	0.656	2.715 ^b^	28.000	145.000	0.000	0.344	1.000
Gender* Area variables	0.890	0.642 ^b^	28.000	145.000	0.915	0.110	0.559
Gender* Age stages	0.878	0.721 ^b^	28.000	145.000	0.844	0.122	0.626
Area variables* Age stages	0.739	1.825 ^b^	28.000	145.000	0.012	0.261	0.990
Gender* Area variables* Age stages	0.863	0.825 ^b^	28.000	145.000	0.718	0.137	0.706

a. Design: Gender + Area variables + Age stages + Gender* Area variables + Gender* Age stages + Area variables* Age stages + Gender* Area variables* Age stages; b. Exact statistic; λ—Wilk’s lambda; F—Fisher test; df—degrees of freedom; Sig.—level of probability; η^2^_p_—partial eta squared.

**Table 5 ijerph-18-12577-t005:** The results of univariate tests/ANOVA and the comparison in pairs of the average values at the level of the items for factor 2/leisure limiting factors.

Dependent Variable	Group	Mean ± Std. Error	a–b	F (1172)	Sig. ^b^	η^2^_p_	Observed Power
F2.1c Limitation/household activities	a. male	2.589 ± 0.197	−0.777 *	8.840	0.003	0.049	0.841
	b. female	3.366 ± 0.172					
F2.2a Stress/going out with friends	a. male	1.553 ± 0.127	0.357 *	4.517	0.035	0.026	0.561
	b. female	1.195 ± 0.111					
F2.2b Stress/reading	a. male	2.137 ± 0.207	0.605 *	4.850	0.029	0.027	0.591
	b. female	1.532 ± 0.181					
F2.2c Stress/listening to music	a. male	1.728 ± 0.139	0.600 *	10.500	0.001	0.058	0.897
	b. female	1.128 ± 0.122					
F2.2f Stress/walks in the park	a. male	1.490 ± 0.092	0.274 *	4.996	0.027	0.028	0.604
	b. female	1.216 ± 0.081					
F2.1d Limitation/commute	a. urban	1.845 ± 0.125	−1.034 *	12.728	0.000	0.069	0.944
	b. rural	2.878 ± 0.261					
F2.2e Stress/various sports activities	a. urban	1.255 ± 0.080	−0.507 *	7.550	0.007	0.042	0.780
	b. rural	1.762 ± 0.166					
F2.2f Stress/walks in the park	a. urban	1.178 ± 0.053	−0.350 *	8.180	0.005	0.045	0.812
	b. rural	1.528 ± 0.110					
F2.2h Stress/socializing on the internet	a. urban	1.746 ± 0.091	−0.676 *	10.275	0.002	0.056	0.890
	b. rural	2.422 ± 0.190					
F2.1a Limitation/working overtime	a. <25 years	1.841 ± 0.098	−0.691 *	5.371	0.022	0.030	0.635
	b. >25 years	2.532 ± 0.282					
F2.1c Limitation/household activities	a. <25 years	2.594 ± 0.086	−0.769 *	8.649	0.004	0.048	0.833
	b. >25 years	3.362 ± 0.247					
F2.1d Limitation/commute	a. <25 years	1.973 ± 0.096	−0.777 *	7.192	0.008	0.040	0.760
	b. >25 years	2.750 ± 0.274					
F2.2b Stress/reading	a. <25 years	2.125 ± 0.091	0.582 *	4.490	0.036	0.025	0.559
	b. >25 years	1.543 ± 0.259					
F2.2f Stress/walks in the park	a. <25 years	1.174 ± 0.040	−0.358 *	8.531	0.004	0.047	0.828
	b. >25 years	1.532 ± 0.116					
F2.2h Stress/socializing on the internet	a. <25 years	1.584 ± 0.070	−0.999 *	22.455	0.000	0.115	0.997
	b. >25 years	2.583 ± 0.199					
F2.2k Stress/shopping	a. <25 years	1.984 ± 0.088	−0.547 *	4.161	0.043	0.024	0.527
	b. >25 years	2.530 ± 0.253					

*. The mean difference is significant at the 0.05 level. ^b^. Adjustment for multiple comparisons: Bonferroni.

**Table 6 ijerph-18-12577-t006:** The results of the Multivariate Tests (MANOVA)/F3—(preferred leisure activities).

Effect	λ	F	Hypothesis df	Error df	Sig.	η^2^_p_	Observed Power
Gender	0.829	0.910 ^b^	32.000	141.000	0.609	0.171	0.797
Area variables	0.829	0.910 ^b^	32.000	141.000	0.609	0.171	0.797
Age stages	0.699	1.901 ^b^	32.000	141.000	0.006	0.301	0.996
Gender* Area variables	0.869	0.663 ^b^	32.000	141.000	0.913	0.131	0.611
Gender* Age stages	0.887	0.560 ^b^	32.000	141.000	0.971	0.113	0.514
Area variables* Age stages	0.847	0.799 ^b^	32.000	141.000	0.768	0.153	0.723
Gender* Area variables* Age stages	0.795	1.136 ^b^	32.000	141.000	0.300	0.205	0.901

Design: Gender + Area variables + Age stages + Gender* Area variables + Gender* Age stages + Area variables* Age stages + Gender* Area variables* Age stages; ^b^. Exact statistic; λ—Wilk’s lambda; F—Fisher test; df—degrees of freedom; Sig.—level of probability; η^2^_p_—partial eta squared.

**Table 7 ijerph-18-12577-t007:** The results of univariate tests/ANOVA and the comparison in pairs of the average values at the level of the items for factor 3 (preferred leisure activities).

Dependent Variable	Group	Mean ± Std. Error	a–b	F (1172)	Sig. ^b^	η^2^_p_	Observed Power
F3.2b Daily activities/reading	a. male	2.203 ± 0.180	−0.526 *	4.837	0.029	0.027	0.590
	b. female	2.728 ± 0.157					
F3.2e Daily activities/various sports activities	a. male	3.588 ± 0.190	0.608 *	5.838	0.017	0.033	0.671
	b. female	2.980 ± 0.165					
F3.2g Daily activities/visits to relatives, friends	a. male	2.088 ± 0.166	−0.472 *	4.594	0.033	0.026	0.568
	b. female	2.559 ± 0.145					
F3.3b Weekend activities/reading	a. male	2.146 ± 0.174	−0.464 *	3.979	0.048	0.023	0.510
	b. female	2.610 ± 0.153					
F3.3e Weekend activities/various sports activities	a. male	3.535 ± 0.187	0.678 *	7.489	0.007	0.042	0.777
	b. female	2.858 ± 0.163					
F3.4a Domestic holidays at sea	a. urban	3.354 ± 0.133	0.612 *	3.949	0.048	0.022	0.506
	b. rural	2.742 ± 0.278					
F3.1a Spending free time/family members	a. <25 years	3.395 ± 0.082	−0.969 *	15.101	0.000	0.081	0.972
	b. >25 years	4.364 ± 0.235					
F3.1b Spending free time/schoolmates or peers	a. <25 years	3.415 ± 0.072	0.783 *	13.003	0.000	0.070	0.948
	b. >25 years	2.631 ± 0.205					
F3.2a Daily activities/going out with friends	a. <25 years	3.068 ± 0.075	0.499 *	4.768	0.030	0.027	0.584
	b. > 25years	2.569 ± 0.216					
F3.2e Daily activities/various sports activities	a. <25 years	3.619 ± 0.083	0.671 *	7.106	0.008	0.040	0.755
	b. >25 years	2.949 ± 0.238					
F3.2h Daily activities/socializing on the internet	a. <25 years	3.739 ± 0.080	0.821 *	11.476	0.001	0.063	0.921
	b. >25 years	2.918 ± 0.229					
F3.3a Weekend activities/going out with friends	a. <25 years	3.593 ± 0.074	0.741 *	11.035	0.001	0.060	0.910
	b. >25 years	2.853 ± 0.210					
F3.3h Weekend activities/socializing on the internet	a. <25 years	3.758 ± 0.083	0.737 *	8.515	0.004	0.047	0.827
	b. >25 years	3.021 ± 0.239					

*. The mean difference is significant at the 0.05 level; ^b^. Adjustment for multiple comparisons: Bonferroni.

**Table 8 ijerph-18-12577-t008:** The results of the Multivariate Tests (MANOVA)/F4 (leisure sports activities).

Effect	λ	F	Hypothesis df	Error df	Sig.	η^2^_p_	Observed Power
Gender	0.790	2.288 ^b^	18.000	155.000	0.003	0.210	0.988
Area variables	0.913	0.821 ^b^	18.000	155.000	0.674	0.087	0.576
Age stages	0.894	1.021 ^b^	18.000	155.000	0.440	0.106	0.701
Gender* Area variables	0.965	0.311 ^b^	18.000	155.000	0.997	0.035	0.208
Gender* Age stages	0.922	0.728 ^b^	18.000	155.000	0.778	0.078	0.511
Area variables* Age stages	0.922	0.730 ^b^	18.000	155.000	0.776	0.078	0.513
Gender* Area variables* Age stages	0.943	0.517 ^b^	18.000	155.000	0.947	0.057	0.355

Design: Gender + Area variables + Age stages + Gender* Area variables + Gender* Age stages + Area variables* Age stages + Gender* Area variables* Age stages; b. Exact statistic; λ—Wilk’s lambda; F—Fisher test; df—degrees of freedom; Sig.—level of probability; η^2^_p_—partial eta squared.

**Table 9 ijerph-18-12577-t009:** The results of the univariate tests/ANOVA and the comparison in pairs of the average values at the level of the items for factor 4 (leisure sports activities).

Dependent Variable	Group	Mean ± Std. Error	a–b	F (1172)	Sig. ^b^	η^2^_p_	Observed Power
F4.4 Satisfaction produced by physical effort	a. male	3.975 ± 0.146	0.406 *	4.385	0.038	0.025	0.549
	b. female	3.569 ± 0.128					
F4.5a Practice/Sports games	a. male	3.403 ± 0.220	0.919 *	9.905	0.002	0.054	0.879
	b. female	2.484 ± 0.192					
F4.5d Practice/Tennis or table tennis	a. male	2.460 ± 0.198	0.831 *	10.011	0.002	0.055	0.882
	b. female	1.629 ± 0.173					
F4.5g Practice/cycling, rollerblading	a. male	2.213 ± 0.190	−0.505 *	4.015	0.047	0.023	0.513
	b. female	2.718 ± 0.166					
F4.5b Practice/Jogging	a. urban	2.939 ± 0.113	0.561 *	4.585	0.034	0.026	0.567
	b. rural	2.378 ± 0.236					
F4.1 Active lifestyle	a. <25 years	3.606 ± 0.077	0.533 *	5.233	0.023	0.030	0.624
	b. >25 years	3.074 ± 0.220					
	b. >25 years	3.421 ± 0.201					

*. The mean difference is significant at the 0.05 level. ^b^. Adjustment for multiple comparisons: Bonferroni.

**Table 10 ijerph-18-12577-t010:** Factor summary of research results.

F1/Leisure Budget
The fact that over 53% of the surveyed students spent half their free time in passive leisure (TV, computer, internet, online socialization) shows that current generations are increasingly dependent on technology (as a lifestyle), which is more obvious for young people in urban areas and for those aged <25 years. The free time budget was superior for men and those aged <25 years and these categories were even more satisfied with the organization of their favorite leisure activities. A worrying aspect is the fact that over a quarter of students had an hour or less as time allocated to leisure activities, an aspect generated by multiple causes (the existence of jobs, the need for self-maintenance and family tasks), and only 11% had over 4−6 h of free time available. The possibilities of spending weekend free time outside the place of residence were zero or very weak for 35% of students and the explanations of this behavior were related to the received education, the precarious financial level and convenience.
F2/Leisure Limiting Factors
The main factors limiting the volume of free time referred to domestic activities (with higher scores for women, those in rural areas and those aged >25), working overtime (for those aged >25) and difficult homework/academic tasks. The commute was, of course, a limiting factor for those in rural areas. There was a very short time allocated to helping others, thus highlighting an attitude predominantly focused on meeting their own needs and neglecting the problems of those in the community. It is unpleasantly surprising the high level of stress generated by reading (especially for men and those <25 years old, so young people in this specialization were not attracted by a classic variant of passive leisure), but also by visits to friends/relatives or shopping. Instead, women had more socially oriented behaviors. Those in rural areas were more stressed by the variants of online socialization, internet use and TV programs, being even less familiar with these technologies, even if in recent years they have experienced a strong expansion in rural areas. The financial limitations in engaging in preferred leisure activities were more obvious for those aged <25 and relative to women, so men had higher incomes and young students did not yet have a salary level according to their wishes. Almost 23% of those investigated were severely limited in terms of the financial component, trips and shopping being in the top of hard-to-reach activities.
F3/Preferred Leisure Activities
Leisure time was usually spent with family (especially women, those in rural areas and those aged >25), schoolmates or entourage (men and those <25 years), life partner (those aged >25 years). It should not be neglected, however, that 11% of students said they spent their free time alone or frequently. This phenomenon of social loneliness is a problem that many studies have reported, caused by the expansion of online communication. Favorite activities during the week were socializing on the internet (especially those <25 years old) and practicing different sports (with higher scores for the urban environment, men and those <25 years old), with similar values, but these data were due exclusively to the university specialization of the students. Other studies have note the dominance of forms of passive leisure for young people of the same age. This was followed by going out with friends (especially for those under <25 years old), computer video games and TV (with higher scores for urban) and listening to music (higher values for those under <25 years old). Theater/cinema and reading were neglected, so the cultural concerns of the studied sample were poorly represented. Over the weekend there was a change in behaviors, including going out with friends at the top. There were increases for socializing on the Internet, video games on computer and TV, visits to friends, listening to music and shopping, but unfortunately, they were accompanied by a moderate decline of involvement in physical activities. However, there was also an increase (but weak) in cultural concerns, with women having better scores for reading than men. Almost 44% of students constantly or frequently spent their holidays at home (men more than women, those in rural areas more than those in urban areas) with financial constraints being the main cause, reinforced by the small number of those who could afford holidays abroad (only 11%). Holidays at sea and in the mountains had relatively similar preference scores, but those in urban areas preferred them more.
F4/Leisure Sports Activities
Given the specialization of our sample, the high scores related to the active lifestyle, involvement in sports and satisfaction generated by physical effort cannot represent a surprise, but it is interesting that men and students aged <25 years had higher average values, so these two subcategories were more dedicated to physical activities than women and older students. Sports games led at the top of the preferences, surprisingly followed by fitness options (Bag Boxing, Zumba, Tae Bo, Body Pump, Pilates, Spinning, etc.), strongly promoted in recent years and taken from Western Europe and the US, jogging is also well represented, these variants having higher involvement scores for men. Women had a better score only for cycling/rollerblading, which is also expanding and gaining popularity among young people. Unfortunately, swimming had the lowest practice scores (due to poor logistics/infrastructure at the local level), even if it was at the top in terms of beneficial effects on health and body harmony.

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
