# Peer review of "The Influence of Socio-Demographic Factors on the Forms of Leisure for the Students at the Faculty of Physical Education and Sports"

_ijerph, 2021, doi:10.3390/ijerph182312577_

Round 1

Reviewer 1 Report

The indications made have been considered. So the article could be accepted for publication.

Author Response

Dear Reviewer,

Thank you for your favorable evaluation and for your patience in reading this material.  

Respectfully,

Correspondent author,

Professor Gabriel Murariu.

22.11.2021

Reviewer 2 Report

The study about the influence of socio-demographic factors on leisure activates among students. The topic is interesting and important, but the description of the methods needs extended. The readers can't get enough information to understand the measures of the study. Without this information, the results are hard to understand.

I have the following suggestions for the authors:  

  • Please use a new chapter for the "purpose of the study" and "hypotheses" or add to the end of the "instruction."
  • There is a lack of information about the measures. Please add more details on the used scales. I suggest reorganizing in different subsections (e.g., leisure activities: write all necessary information about these measures and explain the factors you are using). The readers should get all the important information about the methods in this section.
  • Please introduce those variables in the methods that you analyze.
  • Cronbach alpha is a reliability value (line 245). Please use this term.
  • I recommend introducing all variables in one table. Include: mean, SD, Cronbach alpha
  • One of the main problems of the few information about the measures is that it's not entirely clear is the authors are using factors on their analysis or single questions.

Author Response

Dear Reviewer,

Thank you for the suggested constructive ideas and for the patience to read the evaluated material.

The current changes made to the text are green.

The study about the influence of socio-demographic factors on leisure activates among students. The topic is interesting and important, but the description of the methods needs extended. The readers can't get enough information to understand the measures of the study. Without this information, the results are hard to understand.

I have the following suggestions for the authors:  

  • Please use a new chapter for the "purpose of the study" and "hypotheses" or add to the end of the "instruction."

R: Thank you for the suggestion, the purpose and assumptions were relocated at the end of the introduction.

  • There is a lack of information about the measures. Please add more details on the used scales. I suggest reorganizing in different subsections (e.g., leisure activities: write all necessary information about these measures and explain the factors you are using). The readers should get all the important information about the methods in this section.

R: Thank you for the indication, the Likert scale for all items of the questionnaire is available in the annexes of this study. We present a table that quantifies the intensity of the answers on the 5 steps, but in the paper we cannot transfer it, as this questionnaire was also used to measure the answers in another paper: http://www.gymnasium.ub.ro/index.php/journal/article/view/596/774) and we want to avoid autoplagiarism.  However, in the text they will allocate a paragraph in which we will selectively introduce some of the qualifications related to the 5 levels of intensity.

Table - Distribution of the intensity of the response options, according to the score related to the Likert scale

5

4

3

2

1

Very high

High

Medium

Low

Very low

Very important

Important

Medium importance

Less important

Lacking importance

All the time

Almost all the time

Half of the time

Very little time

Scarcely

Constant

Often/frequent

Occasionally 

Rarely 

Never

Very much

Much

Average

Little

Too little / Not at all

Very satisfied

Fairly satisfied

Medium satisfied

Slightly dissatisfied

Totally dissatisfied

Extreme stress

Strong stress

Medium stress

Reduced stress

Lack of stress

Inaccessible

Hardly accessible

Medium accessible

Easily accessible

Extremely easy

Extremely athletic

Frequently based on physical effort

Moderate involvement in physical effort

Occasional physical effort

Sedentary

Strong satisfaction

Well-being

Moderate comfort

Low comfort and satisfaction

Pain and discomfort

Very strong influence

Strong influence

Medium influence

Weak influence

Lack of influence

  • Please introduce those variables in the methods that you analyze.

R.: Thank you for the indication. We will redo the paragraph that explains the relationship between independent variables (gender, environment and age stages) on the independent variables represented by the items of the questionnaire. (The study was based on determining the influence of the 3 independent variables (gender, area and age stages) and the interaction between them on the dependent variables represented by the an-swers to the questionnaire items.)

  • Cronbach alpha is a reliability value (line 245). Please use this term.

R.: Thank you for the suggestion, the two terms are synonymous. The name in the text and the name of table 1 have been changed.

  • I recommend introducing all variables in one table. Include: mean, SD, Cronbach alpha

R.: Thank you for the suggestion, but the average values are already shown in the 4 tables in the annexes, differentiated according to the items associated with each factor. In the article, only the items for which there are statistically significant differences are registered for each factor. Cronbach alpha values are calculated separately for the 4 factors, not for each item. The academic editor's recommendation was to remove as many tables as possible, to retain the average values for the 85 items and sub-items involves a huge volume of data, which would load the article with at least 5 pages.

  • One of the main problems of the few information about the measures is that it's not entirely clear is the authors are using factors on their analysis or single questions.

R.: Thank you for the question. We used items with closed answers, grouped on the 4 listed factors: F1 / leisure budge, F2 / leisure limiting factors, F3 / preferred leisure activities, F4 / leisure sports activities. The coding of the items produced the ambiguity: F1.3 means item 3 of factor 1, F4.2 means item 2 of factor 4, etc. For an in-depth understanding of the study, we recommend that readers to consult the attached appendices, which could not be included in the article.

Submission Date

28 October 2021

Date of this review

03 Nov 2021 14:36:55

Respectfully,

Correspondent author,

Professor Gabriel Murariu.

22.11.2021

Reviewer 3 Report

I suggest to include qualitative table with resume of the results. I find too much numbers and statistic references and less attention to qualitative data, significance, the possibility of the use of obtained data in the organization of education or for social psychology. Please, include some theoretical data or background useful for the study. The authors should pay more attention to theoretical aspect, qualitative results and possible use of the data.

Author Response

Dear Reviewer,

Thank you for the suggested constructive ideas and for the patience to read the evaluated material.

In the attached table we tried to summarize the essential aspects of our study for the 4 analyzed factors, information that can provide a fairly objective picture of leisure for students of the Faculty of Physical Education and Sports at the University of the Lower Danube in Galati. They can serve as a basis for comparison for the topic addressed, for new demarcated studies and possible changes in the education system, to provide solutions to the problems identified. However, we have included some percentage values, in order to give a scientific support to the presented data. In the paper this table appears in green font, after the results.

The current changes made to the text are green.

Respectfully,

Correspondent author,

Professor Gabriel Murariu.

22.11.2021

Table – Factor summary of research results

F1/leisure budget

·   The fact that over 53% of the surveyed students spend half their free time in passive leisure (TV, computer, internet, online socialization) shows that current generations are increasingly dependent on technology (as a lifestyle), which is more obvious for young people in urban areas and for those aged <25 years.

·   The free time budget is superior to men and those aged <25 years, these categories being even more satisfied with the way of organizing their favorite leisure activities.

·   A worrying aspect is the fact that over a quarter of students have an hour or less as time allocated to leisure activities, an aspect generated by multiple causes (the existence of jobs, the need for self-maintenance and family tasks), and only 11% have over 4 -6 hours of free time available.

·   The possibilities of spending weekend free time outside the place of residence are zero or very weak for 35% of students, the explanations of this behavior being related to the received education, the precarious financial level and convenience.

F2/leisure limiting factors

·   The main factors limiting the volume of free time refer to domestic activities (with higher scores for women, those in rural areas and those aged> 25), working overtime (for those aged> 25) and difficult homework / academic tasks. The commute is, of course, a limiting factor for those in rural areas. There is a very short time allocated to helping others, thus highlighting an attitude predominantly focused on meeting their own needs and neglecting the problems of those in the community.

·   It is unpleasantly surprising the high level of stress generated by reading (especially for men and those <25 years old, so young people in this specialization are not attracted by a classic variant of passive leisure), but also by visits to friends / relatives or shopping, women instead having more socially oriented behaviors.

·   Those in rural areas are more stressed by the variants of online socialization, internet use and TV programs, being even less familiar with these technologies, even if in recent years they have experienced a strong expansion in rural areas.

·   The financial limitations in engaging in preferred leisure activities are more obvious for those aged <25 and relative to women, so men have higher incomes, young students do not yet have a salary level according to their wishes. Almost 23% of those investigated are severely limited in terms of the financial component, trips and shopping being in the top of hard-to-reach activities.

F3/preferred leisure activities 

·   Leisure time is usually spent with family (especially women, those in rural areas and those aged> 25), schoolmates or entourage (men and those <25 years), life partner (those aged > 25 years). It should not be neglected, however, that 11% of students say they spend their free time alone or frequently, this phenomenon of social loneliness being a problem that many studies have reported, being caused by the expansion of online communication.

·   Favorite activities during the week are socializing on the internet (especially those <25 years old) and practicing different sports (with higher scores for the urban environment, men and those <25 years old), with similar values, but these data are due exclusively to the university specialization of students, other studies noting the dominance of forms of passive leisure for young people of the same age. This is followed by going out with friends (especially for those under <25 years old), computer video games and TV (with higher scores for urban) and listening to music (higher values for those under <25 years old). Theater / cinema and reading are neglected, so the cultural concerns of the studied sample are poorly represented.

·   Over the weekend there is a change in behaviors, the tops are going out with friends, there are increases for socializing on the Internet, video games on computer and TV, visits to friends, listening to music and shopping, but unfortunately, they are accompanied by a moderate decline of involvement in physical activities. However, there is also an increase (but weak) in cultural concerns, with women having better scores for reading than men.

·   Almost 44% of students constantly or frequently spend their holidays at home (men more than women, those in rural areas more than those in urban areas), financial constraints being the main cause, reinforced by the small number of those who can afford holidays abroad (only 11%). Holidays at sea and in the mountains have relatively similar preference scores, but those in urban areas prefer them more.

F4/leisure sports activities

·   Given the specialization of our sample, the high scores related to the active lifestyle, involvement in sports and satisfaction generated by physical effort cannot represent a surprise, but it is interesting that men and students aged <25 years have higher average values, so these 2 subcategories are more dedicated to physical activities than women and older students.

·   Sports games lead in the top of the preferences, surprisingly followed by fitness options (Bag Boxing, Zumba, Tae Bo, Body Pump, Pilates, Spinning, etc.), strongly promoted in recent years and taken from Western Europe and the US, jogging is also well represented, these variants having higher involvement scores for men. Women have a better score only for cycling / rollerblading, which is also expanding and gaining popularity among young people. Unfortunately, swimming has the lowest practice scores (due to poor logistics / infrastructure at the local level), even if it is at the top in terms of beneficial effects on health and body harmony.

Respectfully,

Correspondent author,

Professor Gabriel Murariu.

22.11.2021

This manuscript is a resubmission of an earlier submission. The following is a list of the peer review reports and author responses from that submission.

Round 1

Reviewer 1 Report

ACCEPT WITH SIGNIFICANT MODIFICATIONS

 The wording of the entire article should be improved. Especially, in: INTRODUCTION, DISCUSSION and CONCLUSIONS.

INTRODUCTION: Sections not related to an inhere coherent exhibition are made. It does not provide an adequate view of the status of the problem to be investigated.

IT MUST BE RECONSIDERED. METHOD. PARTICIPANTS: It is not indicated how the participants were selected.

COMPLETE THIS SECTION. RESULTS. Very extensive tables. There are difficulties in understanding the results obtained.

CLARIFY THIS SECTION CONCLUSIONS. VERY INCOMPLETE AND LITTLE RELATED TO THE RESULTS OBTAINED. RETHINK.

ACCEPT WITH SIGNIFICANT MODIFICATIONS

 The wording of the entire article should be improved. Especially, in: INTRODUCTION, DISCUSSION and CONCLUSIONS.

INTRODUCTION: Sections not related to an inhere coherent exhibition are made. It does not provide an adequate view of the status of the problem to be investigated.

IT MUST BE RECONSIDERED. METHOD. PARTICIPANTS: It is not indicated how the participants were selected.

COMPLETE THIS SECTION. RESULTS. Very extensive tables. There are difficulties in understanding the results obtained.

CLARIFY THIS SECTION CONCLUSIONS. VERY INCOMPLETE AND LITTLE RELATED TO THE RESULTS OBTAINED. RETHINK.

Reviewer 2 Report

Thank you for the author's contribution of the manuscript entitled: "The influence of gender, the area of origin and the age stages on the forms of leisure for the students of the Faculty of Physical Education and Sports" The manuscript deals with an important topic, but the manuscript need to improve. At first, it's not clear that the authors want to introduce a new measurement for leisure activity, or they would like to introduce the influence of sociodemographic on students. Furthermore, the study's main purpose is not well presented, and the methods are missing important information, which would make the results more understandable.

My detailed comments:

  • In the introduction, there is a lot of information among different countries; however, the study focuses only on the faculty of physical education and sports students. I suggest focusing the introduction on this topic and use only the relevant literature. 
  • Title recommendation: The influence of Sociodemographic on...
  • Intext citations: Please proved name in specific cases (e.g., 77)
  • The purpose of the study and hypotheses should be in the "introduction." 
  • These hypotheses would be suitable for a validation study, but this is not this kind of study. 
  • Please rename: "Research organization" to measures.
  • Measures should be more specific and more detailed. For example, it is not clear how "preferred leisure activates" were measured with a Likert-type scale. 
  • It is very hard to understand the results without a proper introduction of the measures, but I need to acknowledge that the Cronbach alphas are relatively low since the authors used many items. 
  • If the authors would like to introduce a new measure of leisure activity, I recommend using factor analysis to see the factor structure of the questionnaire. 
  • There is not clear why the authors choose ANOVA in Table 3, 5, 7, 9 instead of independent sample T-test.
  • The manuscript needs an English proofread.